# A Novel Procedure for the Management of Severe Hyphema after Glaucoma Filtering Surgery: Air–Blood Exchange under a Slit-Lamp Biomicroscopy

**DOI:** 10.3390/medicina57080855

**Published:** 2021-08-22

**Authors:** I-Hung Lin, Lung-Chi Lee, Ke-Hao Huang, Chang-Min Liang, Yi-Hao Chen, Da-Wen Lu

**Affiliations:** 1Department of Ophthalmology, Tri-Service General Hospital, National Defense Medical Center, Taipei 11490, Taiwan; petercard@gmail.com (I.-H.L.); kidday0205@gmail.com (L.-C.L.); a912572000@gmail.com (K.-H.H.); doc30875@yahoo.com.tw (C.-M.L.); 2Department of Medicine, National Defense Medical Center, Taipei 11490, Taiwan

**Keywords:** filtering surgery, hyphema, office-based, slit-lamp, trabeculectomy

## Abstract

*Background and Objectives*: This study introduces a novel office-based procedure involving air–blood exchange under a slit-lamp microscope for treatment of severe hyphema after filtering surgery. *Materials and Methods*: This retrospective study enrolled 17 patients (17 eyes) with a diagnosis of primary open-angle glaucoma with severe hyphema (≥4-mm height) after filtering surgery. All patients were treated with air–blood exchange under a slit-lamp using room air (12 patients) or 12% perfluoropropane (C3F8; five patients). *Results*: The procedures were successful in all 17 patients; they exhibited clear visual axes without complications during follow-up. In the room air group, the mean visual acuity (VA) and hyphema height significantly improved from 1.70 ± 1.07 LogMAR and 5.75 ± 1.14 mm before the procedure to 0.67 ± 0.18 LogMAR and 2.83 ± 0.54 mm after the procedure (*p* = 0.004; *p* < 0.001). In the C3F8 group, the mean VA showed a trend, though not significant, for improvement from 1.70 ± 1.10 LogMAR to 0.70 ± 0.19 LogMAR (*p* = 0.08); the mean hyphema height showed a trend for improvement from 5.40 ± 0.96 mm to 3.30 ± 0.45 mm. Compared with the C3F8 group, the room air group showed the same efficacy with a shorter VA recovery time. *Conclusions*: “Air–blood exchange under a slit-lamp using room air” is a convenient, rapid, inexpensive, and effective treatment option for severe hyphema after filtering surgery, and may reduce the risk of failure of filtering surgery.

## 1. Introduction

Hyphema is defined as the accumulation of red blood cells in the anterior chamber of the eye; the blood must be grossly visible, either on direct inspection or via slit-lamp examination [1]. This blood accumulation is a result of the disruption of the vessels at the root of the iris or anterior face of the ciliary body, typically due to trauma, underlying medical conditions, or surgical complications [2]. Hyphema is a common complication that can occur after glaucoma filtering surgery [3,4], with a reported incidence of 3.9% to 56% [5,6,7,8,9]. Hyphema management may range from simple/conservative (such as observation using a patch and shield, cycloplegic agents, systemic or topical steroid drops, antifibrinolytic agents, analgesics, and antiglaucoma medications) to surgical interventions (anterior chamber washout with clot removal) [10]. The usual management of the hyphema after filtering surgery is observation as a conservative treatment. However, in some cases, the hyphema may persist or even become worse during observation. Persistent hyphema may cause delay of visual recovery, and therefore further induce anxiety in patients [11]. Surgical interventions (anterior chamber washout with clot removal) can be a successful and easy solution for these patients [10], albeit costly and inconvenient. In particular, non-urgent surgical interventions were hard to perform during the coronavirus disease 2019 (COVID-19) pandemic, starting in 2019, because of delays stemming from government policies and the increased risk of infection. In this study, we introduce a novel ambulatory procedure for the treatment of severe hyphema after filtering surgery, that involves air–blood exchange under a slit-lamp microscope. Our procedure may help patients obtain prompt visual recovery, reduce their anxiety, and reduce the risk of failure of filtering surgery, without the cost and risks inherent to complex surgical procedures and hospital visits in a more convenient, faster, and less expensive way than conventional surgical management.

## 2. Materials and Methods

### 2.1. Patients

This was a retrospective study of 17 eyes of 17 patients treated with our novel procedure at Tri-Service General Hospital, National Defense Medical Center, Taipei City, Taiwan, from January 2017 to January 2021. Our novel treatment procedure, also referred to as “air–blood exchange under a slit-lamp”, was performed on all 17 patients (room air was used for 12 patients and 12% perfluoropropane (C3F8) was used for five patients), who exhibited an obstruction of the visual axis because of severe hyphema after filtering surgery on postoperative day 1; after observation for a week, severe hyphema was still noted or worse at one week postoperative (Figure 1a). 

The inclusion criteria were as follows: a definite diagnosis of primary open-angle glaucoma, and severe hyphema with a height of ≥4 mm after filtering surgery on postoperative one week. The cause of hyphema was nonspecific. Patients with a history of any other ocular surgery or a history of ocular trauma before the filtering surgery were excluded. 

The study was conducted according to the guidelines of the Declaration of Helsinki and approved by the Institutional Review Board of Tri-Service General Hospital, National Defense Medical Center (Registration No. B202005151). All patients were appropriately informed before their participation in the novel procedure and provided written informed consent in accordance with institutional guidelines.

### 2.2. Patient Involvement 

This research was a retrospective study conducted by chart review without patient involvement. Patients were not invited to comment on the study design and were not consulted to develop patient-relevant outcomes or interpret the results. Patients were not invited to contribute to the writing or editing of this document for readability or accuracy.

### 2.3. Treatment Procedure

The treatment procedure was performed by experienced ophthalmologists in Tri-Service General Hospital. These ophthalmologists had completed many difficult anterior chamber paracentesis procedures under a slit-lamp on patients with acute primary angle closure with shallow chamber, with very few and mild complications [12]. The details of the treatment procedure are as follows: 

Step 2: We inserted a 1-cc syringe (filled with room air or 12% C3F8 through the filter (Millex^®^-GV Syringe Filter Unit, 0.22 µm; Merck KGaA, Darmstadt, Germany) for sterilization) with a 25-gauge needle through the inferotemporal limbal cornea of the left eye or inferonasal limbal cornea of the right eye, as close to the 4 o'clock position as possible, under a slit-lamp biomicroscope. We then injected room air or 12% C3F8 through the syringe (Figure 1c and Figure 2a). 

Step 1: We positioned the patient on an office examination chair with the head on the slit-lamp. After applying povidone-iodine solution and administering local anesthesia with proparacaine hydrochloride ophthalmic solution (Alcaine^®^ 0.5%, Alcon Laboratories, Fort Worth, TX, USA), the upper and lower eyelids were held in place with cotton wool swabs by an assistant while the patient’s head was leant and fixed on the forehead rest by an assistant’s hand. We used a 25-gauge needle to stab and make a horizontal extension to create a 1-mm clear cornea incision at the 6 o'clock position of limbal cornea (Figure 1b).

Step 3: The syringe was then removed from the inferotemporal limbal cornea of the left eye or the inferonasal limbal cornea of the right eye. Next, we used a cotton wool swab to gently push around the 1-mm clear cornea incision at the 6 o'clock position under the slit-lamp. As the hyphema was located near the 6 o'clock position due to gravity, the blood in the anterior chamber was then drained from the clear cornea incision (Figure 1d and Figure 2b). After the procedure, patients were instructed to sleep with their head elevated on two or three pillows.

With this procedure, most of the hyphema could be drained out, and the anterior chamber was filled with air bubbles. The visual axis was clear without any blood clots (Figure 1e and Figure 2c). On post-procedure day 2, if any remaining hyphema was noted, we used a cotton wool to swab gently push around the 1-mm clear cornea incision at the 6 o'clock position under the slit-lamp, to let the remaining hyphema to drain from the clear cornea incision. We call this auxiliary procedure on post-procedure day 2 “cotton wool swab pushing”. Tobramycin/dexamethasone (Tobradex®, Alcon Laboratories, Fort Worth, TX, USA) ophthalmic solution was prescribed to be administered every 6 hours for several weeks. 

### 2.4. Treatment Course

In our retrospective study, we performed our novel treatment procedure with room air for 12 patients and 12% C3F8 for five patients. All patients presented with hyphema after filtering surgery on postoperative day 1, which we observed for one week; however, severe hyphema (≥4 mm) was still noted on postoperative day 7. Then, all patients underwent room air–blood exchange under a slit-lamp on postoperative day 7. On postoperative day 8, all patients presented with lower hyphema relative to that on postoperative day 7. Due to the lower quantity of remaining hyphema, we performed the cotton wool swab pushing on postoperative day 8, to let the remaining hyphema drain from the clear cornea incision. We checked the visual acuity (VA), intraocular pressure (IOP), and height of hyphema before the procedure on postoperative day 7, after the procedure but before the cotton wool swab pushing on postoperative day 8, and 1 and 2 weeks after the procedure. We also checked the height of air bubbles in the anterior chamber before the cotton wool swab pushing on postoperative day 8 and at 1 and 2 weeks after the procedure. All patients underwent a standard ophthalmic examination to determine VA and slit-lamp biomicroscopy to examine the anterior segment, including measurement of the height of hyphema and air bubbles. Goldmann applanation tonometry was used to determine the IOP. The follow-up period after filtration surgery was 3 months for all patients. 

### 2.5. Statistical Analysis

Excel version 2019 for Windows (Microsoft Corp., Redmond, WA, USA) was used for statistical analysis. Independent-sample *t*-tests were used for the analyses. A *p*-value of <0.05 was considered statistically significant.

## 3. Results

For the room air group 12 patients (five men and seven women) were included with a mean age of 59.8 ± 8.5 years (range, 47 to 77 years). Regarding underlying medical diseases, four patients had hypertension, three patients had diabetes mellitus, two patients had hypertension combined with diabetes mellitus, and three patients had no specific underlying medical disease. All patients had glaucoma that was surgically managed, either by trabeculectomy (eight patients) or by trabeculectomy combined with phacoemulsification and intraocular lens implantation (four patients) (Table 1). None of them used any anticoagulant medication during the week preceding the filtering surgery. The treatment procedure was successful in all 12 patients, with hyphema clearing within the first week after the procedure. The mean VA significantly improved from 1.70 ± 1.07 LogMAR (range, 0.7 to 2.9) on postoperative day 7, before the procedure, to 0.67 ± 0.18 LogMAR (range, 0.4 to 1.0) on postoperative day 8, after the procedure but before the cotton wool swab pushing (*p* = 0.004). The mean height of hyphema significantly improved from 5.75 ± 1.14 mm (range, 4.5 to 8.0) on postoperative day 7, before the procedure, to 2.83 ± 0.54 mm (range, 2 to 4), on postoperative day 8 after the procedure but before the cotton wool swab pushing (*p* < 0.001). One week after the procedure, the hyphema cleared in all 12 patients. The air bubble in the anterior chamber showed a mean height of 7.33 ± 0.78 mm (range, 6 to 8) on postoperative day 8 after the procedure but before the cotton wool swab pushing, and it cleared in all 12 patients 1 week later. (Table 2). There were no complications related to the procedure in any of the cases. At the 3-month follow-up, there was no recurrence of hyphema.

For the C3F8 group, five patients (two men and three women) were included with a mean age of 59.0 ± 5.4 years (range 51 to 67 years). Regarding underlying medical diseases, two patients had hypertension, one patient had diabetes mellitus, one patient had hypertension combined with diabetes mellitus, and one patient had no specific underlying medical disease. All patients had glaucoma that was surgically managed, either by trabeculectomy (three patients), or trabeculectomy combined with phacoemulsification and intraocular lens implantation (two patients) (Table 3). The treatment procedure was successful in all five patients, although the air bubble remained within the first week and cleared 2 weeks after the procedure. The mean VA showed a trend, though not significant, for improvement from 1.70 ± 1.10 LogMAR (range, 0.7 to 2.9) on postoperative day 7, before the procedure, to 0.70 ± 0.19 LogMAR (range, 0.5 to 1.0) on postoperative day 8, after the procedure, but before the cotton wool swab pushing (*p* = 0.08). The mean height of hyphema showed a trend for improvement from 5.40 ± 0.96 mm (range, 4.5 to 7) on postoperative day 7, before the procedure, to 3.30 ± 0.45 mm (range, 3 to 4) on postoperative day 8, after the procedure, but before the cotton wool swab pushing (*p* = 0.015). One week after the procedure, the hyphema cleared in all five patients. The air bubbles remained within the anterior chamber, with a mean height of 7.40 ± 0.55 mm (range from 7 to 8) on postoperative day 8, after the procedure but before the cotton wool swab pushing. One week after the procedure, the mean height of the air bubbles showed a trend for decrease to 4.80 ± 0.84 mm (range from 4 to 6) (*p* = 0.002). Two weeks later, all air bubbles cleared in all five patients (Table 2). There were no complications related to the procedure in these cases. At the 3-month follow-up, there was no recurrence of hyphema. Although the height of hyphema and air bubbles decreased statistically after C3F8 injection, it must be treated with caution in clinical practice because of the limited case number in our series (*n* = 5). A further study of a large sample size is needed to validate significance.

## 4. Discussion

Currently, most patients undergo surgical management of severe hyphema [13]. However, anterior chamber paracentesis under the slit-lamp has been proven safe with a complication rate of only 0.7% [14]. It is commonly performed for the diagnostic and therapeutic removal of aqueous fluid in the anterior chamber. Therefore, anterior chamber paracentesis under slit-lamp has the potential to be applied as a rapid, convenient, safe, and inexpensive method for the management of hyphema. 

MomPremier et al. [15] described a new anterior chamber fluid-gas exchange technique for the management of hyphema in 2015. They used retrobulbar anesthesia and a lid speculum. After sterile preparation, filtered gas (12% C3F8 or air) was injected through a 3-cc syringe with a 27-gauge, 1/2-inch needle, which was inserted through the superotemporal limbal cornea (as close to the 12 to 1 o'clock position as possible) into the anterior chamber. At the same time, another tuberculin syringe with the plunger removed and a 1/2-inch, 27-gauge needle was inserted through the inferotemporal limbal cornea (as close to the 5 to 6 o'clock position as possible) into the anterior chamber and the hyphema was drained passively. Compared to this procedure, our technique did not use retrobulbar anesthesia or a lid speculum. We used local anesthesia with proparacaine hydrochloride solution 0.5%. Moreover, we used one syringe instead of two. Therefore, our technique is more convenient and less expensive than the technique described by MomPremier et al. In all 17 cases, most of the hyphema was drained successfully after performing the procedure one week after filtering surgery. VA and the height of hyphema improved immediately after performing the procedure (Figure 3 and Figure 4; Table 3), and the patients’ hyphema cleared completely within 1 additional week (Table 3). None of the patients were required to undergo reoperation for further treatment of the hyphema, and no other complications were noted during the 3-month follow-up. 

The hyphema was successfully drained with an improvement in VA in both groups; however, the length of time for which the air bubble remained in the anterior chamber differed between groups. In the room air group, the air bubble cleared in all 12 patients 1 week after the procedure. In contrast, in the C3F8 group, the air bubble showed a trend for decrease but remained 1 week after the procedure in all five patients, and it cleared 2 weeks after the procedure (Table 3). Thompson et al. [16] reported that the half-life of room air was 1.3 ± 0.1 days and the half-life of 15% C3F8 was 8.0 ± 0.6 days in the intraocular space. They showed that room air could clear faster than C3F8, which was consistent with the observation in our study. Air bubbles in the anterior chamber can cause blurry vision, which indicates that the use of room air in this novel procedure can result in a shorter VA recovery time relative to the use of C3F8. Furthermore, room air requires no additional financial investment and is easier to obtain than C3F8. Therefore, we believe that this novel procedure is considerably better when performed with room air. We used C3F8 in our first cases, following its use by MomPremier et al. for a similar procedure in 2015 [14]. However, we found after a few cases that using room air in our novel procedure requires could need less VA recovery time and was much easier to obtain than C3F8. Therefore, we adopted room air for procedures thereafter.

The advantages of performing this procedure with room air are as follows: First, hyphema may be a risk factor for the failure of trabeculectomy because it may increase concentrations of some cytokines that may lead to a failure of the conjunctival bleb formation [17]. Therefore, if we can reduce the duration of hyphema, we may also reduce the risk of trabeculectomy failure. Second, relatively long duration of hyphema is a risk factor of corneal blood staining [18]. Therefore, our procedure which can reduces the duration of hyphema may also reduce the risk of corneal blood staining. Third, it is a rapid, convenient, and inexpensive method for the management of hyphema. Conventional surgical intervention in the operation room is more expensive and time consuming. It may also be inconvenient during periods of public health concern, such as the recent global COVID-19 pandemic, since the non-urgent surgery in the operation may be discouraged or delayed due to emergency health policies. Fourth, compared to observation, the procedure with room air may help the patient attain visual recovery earlier and reduce their anxiety about hyphema. Although the air bubbles in the anterior chamber can cause blurry vision, the patients may attain visual recovery earlier after this procedure compared to the conservative management of observation, since the half-life of room air was only 1.3 days as mentioned above. The symptoms of hyphema, including pain, red tint of the visual field, photophobia, blur, clouded or obstructed vision [19] may also cause more anxiety compared with the symptom of blurry vision caused by the air bubbles in the anterior chamber. Finally, for the hemostasis of hyphema, the coagulation processes will proceed well under aqueous humor [20], but because aqueous humor is flowing, it might wash away the clotting factors before a stable clot can form. Therefore, tamponade using air may have a better influence on hemostasis.

Traditionally, ophthalmologists prefer conservative treatment such as observation in cases of hyphema, but our procedure is an alternative treatment for this condition. However, gaining mastery of the technique described herein requires practice. All the ophthalmologists who performed the procedure in the present study have extensive experience in anterior chamber paracentesis under slit-lamp with local anesthesia [12]. Since our 17 cases showed no complications during the 3-month follow-up, the procedure appears to be safe; however, it may be unsafe if attempted by an untrained ophthalmologist. Hence, an ophthalmologist should first train by practicing anterior chamber paracentesis under slit-lamp with local anesthesia in a regular clinical setting. Furthermore, selection of suitable patients is important to perform the procedure successfully. Nervous patients may be injured due to their inability to tolerate the procedure; hence it is better to avoid performing the technique on patients who tend to move their body or eyes during manipulation.

Its viability for managing a clotted hyphema, which may not pass through a 25-gauge needle, may also be limited. However, the clotted hyphema may begin to liquify after observation for one week and thus may offer the chance to be drained with this procedure. Moreover, if the anterior chamber is difficult to see due to severe hyphema or if it is shallow due to the over-filtration caused by the filtering surgery, the procedure will be difficult and dangerous. In this situation, it may be prudent to continue to observe the patient, or perform the surgical interventions (anterior chamber washout with clot removal) in the operation room.

Our study has a few limitations, such as the small sample size, which precludes a statistical treatment in the C3F8 group (*n* = 5), and the short follow-up time. In the future, larger-scale studies with longer follow-up periods will be needed to prove the procedure’s safety and efficacy. Additionally, in our study, the described procedure was useful to wash out the hyphema regardless of the causes. However, the hyphema occurred in patients using anticoagulant medication or having severe coagulopathy may not be resolved by our procedure, and these cases had been excluded in our study. Additional studies are necessary to verify the effect of our procedure on these specific cases. A prospective and controlled study will be helpful for evaluating the effect of our procedure compared with the effect of observation as a conservative treatment. However, this convenient, quick, inexpensive, effective, and unique office-based new technique can still benefit patients with hyphema after filtering surgery. For the majority of patients with hyphema, observation as a conservative treatment is enough. For those with severe hyphema that have no sign of resolution or worsen after observation for one week, or have high anxiety about poor visual acuity due to hyphema, ophthalmologists can try this much more convenient way of managing it before opting for surgical anterior chamber wash out in the operating room. This may help patients have earlier visual recovery, reduce their anxiety, and reduce the risk of failure of filtering surgery. This new procedure may also benefit patients with hyphema following ab interno trabeculotomy or other types of anterior chamber angle surgery because hyphema are very common after these surgeries [21,22], and it may also benefit patients with hyphema from other causes, such as trauma; however, further studies are needed to validate its efficacy.

## 5. Conclusions 

We introduced a novel, office-based treatment procedure called “air–blood exchange under a slit-lamp”, which involves the use of a slit-lamp microscope for the treatment of severe hyphema after filtering surgery. We performed this novel procedure with room air for 12 patients and 12% C3F8 for five patients and showed that the treatment was successful in all patients with a clear visual axis, no complications, and no re-operation. VA recovery can be quicker with the use of room air than with the use of 12% C3F8 in this novel procedure, considering the shorter time for the gas remaining in the anterior chamber. Room air is also cheaper and easier to obtain. Therefore, we consider this procedure using room air to be effective; moreover, it is more convenient, more rapid, and less expensive than conventional surgical management performed in the operating room. Ophthalmologists may consider this treatment option instead of current inconvenient and expensive surgical interventions to help patients obtain visual recovery earlier, reduce their anxiety about hyphema, and may reduce the risk of failure of filtering surgery. 

## Figures and Tables

**Figure 1 medicina-57-00855-f001:**
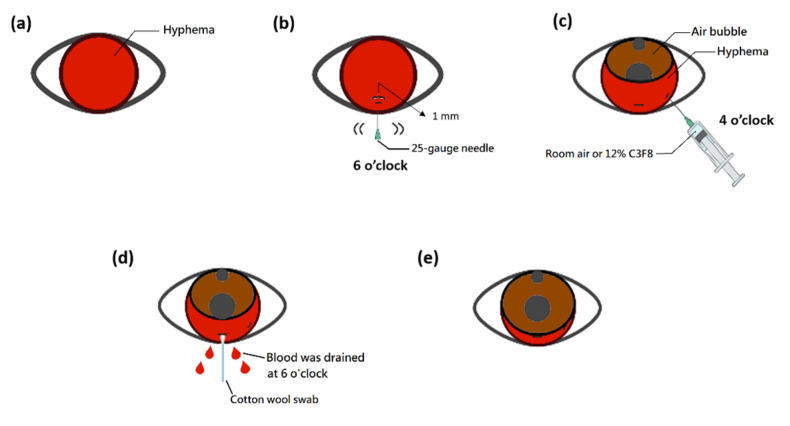
“Air–blood exchange under a slit-lamp’’ for the management of severe hyphema. This diagrammatic depiction was illustrated by one of the co-authors, I-Hung Lin and his colleague, Yi-Han Huang. (**a**) A patient with a blocked visual axis because of severe hyphema after filtering surgery. (**b**) After applying povidone-iodine solution and administering local anesthesia with proparacaine hydrochloride ophthalmic solution (Alcaine® 0.5%, Alcon Laboratories, Fort Worth, TX, USA), the upper and lower eyelids were held in place with cotton wool swabs by an assistant, and the patient’s head was leant and fixed on the forehead rest by an assistant’s hand. We used a 25-gauge needle to stab and make horizontal extension to create a 1-mm clear cornea incision at the 6 o’clock position of limbal cornea. (**c**) A 1-cc syringe with a 25-gauge needle is filled with room air or 12% perfluoropropane (C3F8) and inserted through the inferotemporal limbal cornea of left eye or inferonasal limbal cornea of right eye, as close to the 4 o’clock position as possible, under a slit-lamp biomicroscope. Room air or 12% C3F8 is then injected through the syringe. (**d**) The syringe was then removed from the inferotemporal limbal cornea of the left eye or the inferonasal limbal cornea of the right eye. Then we used a cotton wool swab to gently push around the 1 mm clear cornea incision at 6 o’clock position under the slit-lamp. The blood in the anterior chamber was then drained from the clear cornea incision wound. (**e**) With this procedure, most of the hyphema was drained, and the anterior chamber was filled with air bubbles. The visual axis is clear without any blood clots.

**Figure 2 medicina-57-00855-f002:**
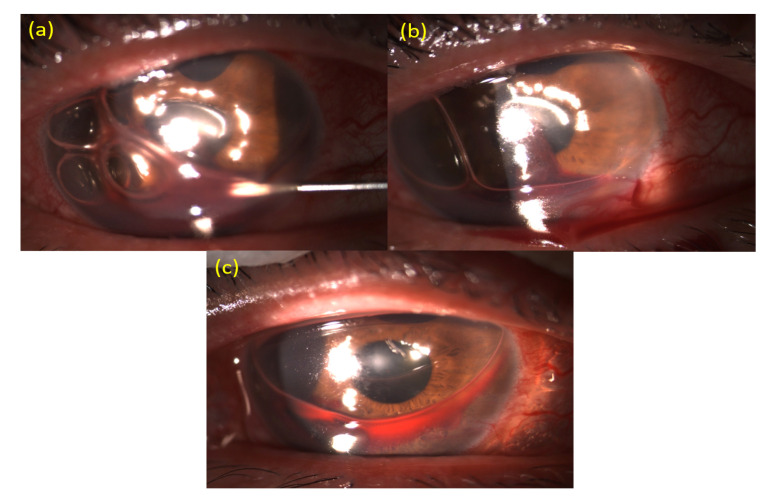
Photos of “air–blood exchange under a slit-lamp’’ for the management of severe hyphema. (**a**) A 1-cc syringe with a 25-gauge needle was filled with room air or 12% perfluoropropane (C3F8) and inserted through the inferotemporal limbal cornea of left eye or inferonasal limbal cornea of right eye, as close to the 4 o'clock position as possible, under a slit-lamp biomicroscope. Room air or 12% C3F8 was then injected through the syringe. (**b**) The syringe was then removed from the inferotemporal limbal cornea of the left eye or the inferonasal limbal cornea of the right eye. We then used a cotton wool swab to gently push around the 1-mm clear cornea incision at the 6 o’clock position under the slit-lamp. The blood in the anterior chamber was then drained from the clear cornea incision; therefore, the blood can be seen outside the cornea at the 6 o'clock position in this figure. (**c**) With the “air–blood exchange” procedure, most of the hyphema was drained, and the anterior chamber is filled with air bubbles. The visual axis is clear without any blood clots.

**Figure 3 medicina-57-00855-f003:**
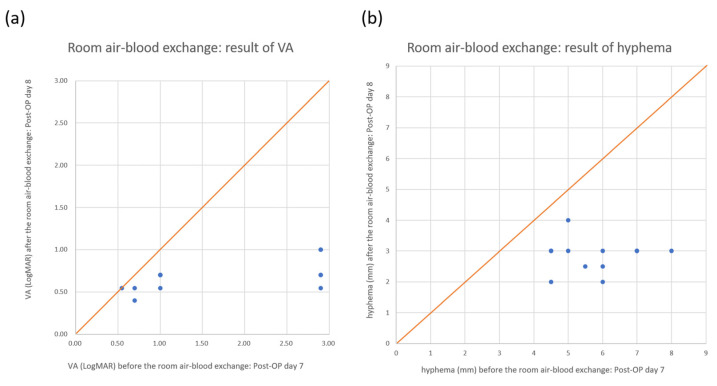
Visual acuity (VA) and hyphema changes after the air–blood exchange procedure with room air. From the scatter plot, we can see that nearly all the spots are below the 45° orange line. This means that VA (**a**) and hyphema (**b**) improved in nearly all the patients.

**Figure 4 medicina-57-00855-f004:**
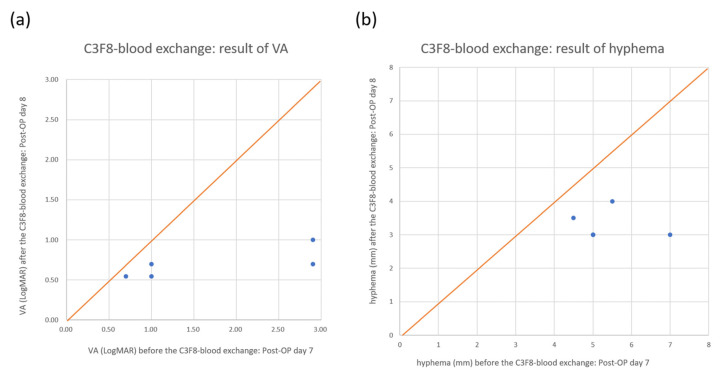
Visual acuity (VA) and hyphema changes after the air–blood exchange procedure with 12% perfluoropropane. From the scatter plot, we can see that all the spots are below the 45° orange line. This means that VA (**a**) and hyphema (**b**) were improved in all the patients.

**Table 1 medicina-57-00855-t001:** Cases with hyphema treated by a novel air–blood exchange procedure using room air following trabeculectomy.

	Age	Sex	Past Medical History	Past OPH Surgery	Post-op Day 7:Before Air–Blood Exchange			Post-op Day 8: After Air–Blood Exchange				Post-Procedure One Week Later		Post-Procedure Two Weeks Later	
					Height of hyphema	VA	IOP	Height of hyphema	VA	IOP	Height of air bubble	Height of hyphema	Height of air bubble	Height of hyphema	Height of air bubble
Case 1	52	F	HTN, DM	TLE + Phaco + IOL	4.5 mm	HM	7	3 mm	20/100	8	7 mm	Clear	Clear	Clear	Clear
Case 2	63	F	HTN	TLE + Phaco + IOL	5 mm	20/200	6	3 mm	20/100	7	8 mm	Clear	Clear	Clear	Clear
Case 3	58	F	DM	TLE + Phaco +IOL	8 mm	HM	6	3 mm	20/200	8	7 mm	Clear	Clear	Clear	Clear
Case 4	62	M	HTN	TLE	4.5 mm	20/200	5	3 mm	20/100	7	8 mm	Clear	Clear	Clear	Clear
Case 5	51	M	DM	TLE	5 mm	20/70	8	4 mm	20/70	8	6 mm	Clear	Clear	Clear	Clear
Case 6	47	F	none	TLE	5.5 mm	20/100	9	2.5 mm	20/70	10	8 mm	Clear	Clear	Clear	Clear
Case 7	71	F	HTN, DM	TLE + Phaco + IOL	6 mm	20/200	8	3 mm	20/100	7	7 mm	Clear	Clear	Clear	Clear
Case 8	60	M	HTN	TLE	4.5 mm	20/100	6	2 mm	20/50	9	8 mm	Clear	Clear	Clear	Clear
Case 9	77	M	DM	TLE + Phaco + IOL	7 mm	HM	10	3 mm	20/70	9	8 mm	Clear	Clear	Clear	Clear
Case 10	57	F	none	TLE	6 mm	20/200	9	2.5 mm	20/70	11	8 mm	Clear	Clear	Clear	Clear
Case 11	64	M	HTN	TLE	6 mm	HM	10	2 mm	20/200	9	7 mm	Clear	Clear	Clear	Clear
Case 12	55	F	none	TLE	7 mm	HM	8	3 mm	20/100	9	6 mm	Clear	Clear	Clear	Clear

DM = diabetes mellitus, HM = hand movement, HTN = hypertension, IOL = intraocular lens implantation, IOP = intraocular pressure, Post-op = postoperative, OPH = ophthalmic, Phaco = phacoemulsification, TLE = trabeculectomy, VA = visual acuity.

**Table 2 medicina-57-00855-t002:** Visual and anatomical outcomes in the room air and perfluoropropane groups.

	Room Air Group	C3F8 Group
BCVA (LogMAR)		
Postoperative days 7: before the procedure	1.70 ± 1.07	1.70 ± 1.10
Postoperative days 8: after the procedure	0.67 ± 0.18	0.70 ± 0.19
*p*-value for the improvement in each group	0.004	0.08
Height of hyphema (mm)		
Postoperative days 7: before the procedure	5.75 ± 1.14	5.40 ± 0.96
Postoperative days 8: after the procedure	2.83 ± 0.54	3.30 ± 0.45
*p*-value for the improvement in each group	<0.001	0.015
Height of air bubble (mm)		
Postoperative days 8: after the procedure	7.33 ± 0.78	7.40 ± 0.55
One week later after the procedure	clear in all patients	4.80 ± 0.84
Two weeks later after the procedure	clear in all patients	clear in all patients

BCVA = best-corrected visual acuity, C3F8 = perfluoropropane.

**Table 3 medicina-57-00855-t003:** Cases with hyphema treated by a novel air–blood exchange procedure using 12% perfluoropropane following trabeculectomy.

	Age	Sex	Past Medical History	Past OPH Surgery	Post-op Days 7: Before Air–Blood Exchange			Post-op Days 8: After Air–Blood Exchange				Post-Procedure One Week Later		Post-procedure Two Weeks Later	
					Height of hyphema	VA	IOP	Height of hyphema	VA	IOP	Height of air bubble	Height of hyphema	Height of air bubble	Height of hyphema	Height of air bubble
Case 1	60	M	HTN	TLE + Phaco + IOL	5 mm	HM	6	3 mm	20/100	7	8 mm	Clear	6 mm	Clear	Clear
Case 2	54	M	HTN	TLE + Phaco + IOL	5.5 mm	20/200	9	4 mm	20/100	9	7 mm	Clear	5 mm	Clear	Clear
Case 3	63	F	HTN, DM	TLE	7 mm	HM	6	3 mm	20/200	8	7 mm	Clear	4 mm	Clear	Clear
Case 4	51	F	None	TLE	4.5 mm	20/200	7	3.5 mm	20/70	8	7 mm	Clear	5 mm	Clear	Clear
Case 5	67	F	DM	TLE	5 mm	20/100	8	3 mm	20/70	9	8 mm	Clear	4 mm	Clear	Clear

DM = diabetes mellitus, HM = hand movement, HTN = hypertension, IOL = intraocular lens implantation, IOP = intraocular pressure, Post-op = postoperative, OPH = ophthalmic, Phaco = phacoemulsification, TLE = trabeculectomy, VA = visual acuity.

## Data Availability

The data used and/or analyzed during the current study are not available for public access because of patient privacy concerns but are available from the corresponding author on reasonable request.

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
