# Peer review of "A Novel Procedure for the Management of Severe Hyphema after Glaucoma Filtering Surgery: Air–Blood Exchange under a Slit-Lamp Biomicroscopy"

_medicina, 2021, doi:10.3390/medicina57080855_

Round 1

Reviewer 1 Report

Congratulations to the Authors of an interesting work.

As mentioned in the discussion, previous reports of procedures for injecting gas or air into the anterior chamber have included retrobulbar anesthesia and slightly different equipment. The reported cases of hyphema were very serious. The height of the blood level in the anterior chamber oscillating around 6 mm (even 8 mm in one case) poses a significant risk of developing further complications. 

The presented work does not describe the diagnostic process of the cause of hyphema in the described cases.

The work raises the following queries:

- Have the causes of hyphema in the presented cases been known?

- Is the described procedure universal, regardless of the cause of the hyphema after trabeculectomy?

- Why did the authors choose this treatment in the following cases? Was the prevailing COVID-19 pandemic the main cause? Cost reduction or medical indications to control hyphemia as soon as possible for fear of later complications?

- In the described method, air / gas is administered in the inferonasal / inferotemporal quadrant (depending on which eye is being treated). What is the reason for selecting a lower location for air / gas injection into the anterior chamber? Is it easier for the operator to access through a slit lamp during the procedure, or for some other reason?

Carrying out the described procedure using a slit lamp with manual maintenance of the eyelids of the patient seems risky. I understand that the Authors would prefer to perform the procedure in an operating room, but the situation forced them to proceed differently.

The work is impressive, very interesting and thought-provoking. The presented cases show that it is an alternative to the waiting attitude that is beneficial for patients

Author Response

Response to Reviewer 1 Comments

Point 1: Have the causes of hyphema in the presented cases been known?

Response 1: Thanks for your question. The causes of hyphema in the presented cases were not known. Just like Mannino et al. has mentioned, the causes of hyphema after glaucoma filtering surgery are not always well known [a]. Also, as we had mentioned at line 218-219, None of them used any anticoagulant medication during the week preceding the filtering surgery. Therefore, anticoagulant medication was not the cause of hyphema. We think the causes of hyphema in our presented cases were nonspecific. We add the information” The cause of hyphema was nonspecific.“ at line 83-84. Thank you very much.

Reference:

  1. Mannino, G., Verrilli, S., Calafiore, S., Ciarnella, A., Cutini, A., Mannino, C., Perdicchi, A., & Recupero, S. M.; Evaluation of recurrent hyphema after trabeculectomy with ultrabiomicroscopy 50-80 MHz: a case report. BMC research notes 2012, 5, 549.

Point 2: Is the described procedure universal, regardless of the cause of the hyphema after trabeculectomy?

Response 2: Thanks for your question. In most of cases, the answer is “Yes.” The described procedure was useful to clean out the hyphema regardless the causes in our study. However, it may be invalid in some specific cases such as patients using anticoagulant medication or having severe coagulopathy. Such cases were not included in our study. A further study is necessary to verify the effect of our procedure on these specific cases. We add a paragraph to discuss this issue at line 407-412.

Line 407-412:

“Besides, in our study, the described procedure was useful to clean out the hyphema regardless the causes. However, the hyphema occurred in patients using anticoagulant medication or having severe coagulopathy may not be resolved by our procedure, and these cases had been excluded in our study. A further study may be necessary to verify the effect of our procedure on these specific cases.”

Point 3: Why did the authors choose this treatment in the following cases? Was the prevailing COVID-19 pandemic the main cause? Cost reduction or medical indications to control hyphemia as soon as possible for fear of later complications?

Response 3: Thanks for your question. The prevailing COVID-19 pandemic was the main cause to drive us to do this procedure. Persistent hyphema may increase concentrations of some cytokines that may lead to a failure of the conjunctival bleb formation. Therefore, if we can reduce the duration of hyphema, we may reduce the risk of trabeculectomy failure. However, the non-urgent surgery in the operation may be discouraged or delayed due to emergency health policies in Taiwan during this COVID-19 pandemic period. Therefore, the rapid and convenient procedure that can be done in the office-based setting was preferred since COVID-19 pandemic began. Of course, even without the COVID-19 pandemic, our procedure was still cheaper and more convenient than that performed in the operative room. We had provided the above discussion at line 354-364.

Point 4: In the described method, air / gas is administered in the inferonasal / inferotemporal quadrant (depending on which eye is being treated). What is the reason for selecting a lower location for air / gas injection into the anterior chamber? Is it easier for the operator to access through a slit lamp during the procedure, or for some other reason?

Response 4: Thanks for your question. As you mentioned, it is easier for the operator to administer the air / gas injection through a slit lamp at a lower location. Although air/ gas injection was done at lower location, it would flow to the upper side of the anterior chamber due to buoyant force, further push the hyphema to drain out at the paracentesis site in the 6 o'clock position.

Reviewer 2 Report

This is a minor variation pf the technique described by Mom Premier M et al. The emphasis regarding the novelty of technique should be therefore reduced. However the scientific interest for these type of office based techniques may be of interest for the readers.

Authors should address the following points:

Risk of pupillary block after air/gas injection and postoperative posture. Are mydriatics recommended? Is an upright posture indicated?

Rationale to choose between filtered air and gas. Is it based on the hyphema height?

Authors are encouraged to consider in the discussion the frequency of hyphema following ab interno trabeculotomy or other types of angle surgery procedures. 

Author Response

Response to Reviewer 2 Comments

Point 1: This is a minor variation pf the technique described by Mom Premier M et al. The emphasis regarding the novelty of technique should be therefore reduced. However the scientific interest for these type of office based techniques may be of interest for the readers.

Response 1: Thanks for your suggestion. We agreed with you. MomPremier et al. used two syringes with 2 needles at the same time. However, his procedure is hardly to do in the office-based setting. In the office-based setting, while one hand was used to hold the syringe, the other hand was needed to control the handle of slit lamp to focus. therefore, we use one syringe with one needle, which can also reduce the risk of damaging the lens or other intraocular structure. Besides, we used local anesthesia to instead the retrobulbar anesthesia and a lid speculum which were used by MomPremier et al. Therefore, our technique is more convenient and less expensive than the technique described by MomPremier et al. We had provided the above information at line 292-307. That’s why we think our technique has some novelty. However, we agreed with you and reduce the regarding phrase of novelty in the part of comparing MomPremier et al.’s and our technique by change” our new technique” to” our technique” at line 302 and 305. If the editor still thought we emphasize too much about the novelty, we can further reduce more regarding phrase. Thank you very much.

Point 2: Risk of pupillary block after air/gas injection and postoperative posture. Are mydriatics recommended? Is an upright posture indicated?

Response 2: As a standard step of trabeculectomy, peripheral iridectomy was done in all the cases. Therefore, the risk of pupillary block was dramatically reduced by this procedure. In our study, there is no pupillary block was noted as complication. Mydriatics were not routinely used in our study. It was used only if shallow chamber was noted after trabeculectomy. For postoperative posture, no special posture was indicated, except that we educated patients to sleep on two or three pillows so their head is elevated when they slept, in order to let the blood cells of hyphema settle to the bottom of the eye, to increase the speed of resolution of the hyphema, and decrease the risk of corneal blood staining. We add the above information at line 129-130 as follows: “After the procedure, we educated patients to sleep on two or three pillows so their head is elevated when they slept.”

Point 3: Rationale to choose between filtered air and gas. Is it based on the hyphema height?

Response 3: It was not based on the hyphema height. At the beginning, we use perfluoropropane (C3F8) in our first cases, following its use by MomPremier et al. for a similar procedure in 2015. However, we found after a few cases that using room air in our novel procedure could need less VA recovery time and was much easier to obtain than C3F8. Therefore, we adopted room air for procedures thereafter. We had provided the above information at line 347-352. Thank you very much.

Point 4: Authors are encouraged to consider in the discussion the frequency of hyphema following ab interno trabeculotomy or other types of angle surgery procedures.

Response 4: Thanks for your suggestion. We add the discussion the frequency of hyphema following ab interno trabeculotomy or other types of angle surgery procedures at line 424-429, and add new references as follows:

“This new procedure may also benefit patients with hyphema following ab interno trabeculotomy or other types of angle surgery because hyphema are very common after these surgeries [20, 21], and it may also benefit patients with hyphema from other causes, such as trauma; however, further studies are needed to further validate its efficacy.”

New references:

  1. Ahuja, Y.; Malihi, M.; Sit, A. J. Delayed-onset symptomatic hyphema after ab interno trabeculotomy surgery. American journal of ophthalmology 2012, 154(3), 476–480.e2.
  2. Minckler, D.; Baerveldt, G.; Ramirez, M. A.; Mosaed, S.; Wilson, R.; Shaarawy, T.; Zack, B.; Dustin, L.; Francis, B. Clinical results with the Trabectome, a novel surgical device for treatment of open-angle glaucoma. Transactions of the American Ophthalmological Society 2006, 104, 40–50.

This manuscript is a resubmission of an earlier submission. The following is a list of the peer review reports and author responses from that submission.